# Chimeric Virus Made from crTMV RNA and the Coat Protein of Potato Leafroll Virus is Targeted to the Nucleolus and Can Infect *Nicotiana benthamiana* Mechanically

**DOI:** 10.3390/ht9020011

**Published:** 2020-04-26

**Authors:** Konstantin O. Butenko, Inna A. Chaban, Eugene V. Skurat, Olga A. Kondakova, Yuri F. Drygin

**Affiliations:** 1Faculty of Biology, Lomonosov Moscow State University, Moscow 119992, Russia; k002@yandex.ru (K.O.B.); skurat@belozersky.msu.ru (E.V.S.); olgakond1@yandex.ru (O.A.K.); 2All-Russian Research Institute of Agricultural Biotechnology, Russian Academy of Agricultural Sciences, Moscow 127422, Russia; inna_chaban@rambler.ru; 3Belozersky Research Institute of Physico-Chemical Biology, Lomonosov Moscow State University, Moscow 119992, Russia

**Keywords:** agroinoculation, binary vector, chimeric virus, recombinant RNA, electron microscopy, mechanical infection of plant, tissue blot-immunoassay

## Abstract

A genetically engineered chimeric virus crTMV-CP-PLRV composed of the crucifer-infecting tobacco mosaic virus (crTMV) RNA and the potato leafroll virus (PLRV) coat protein (CP) was obtained by agroinfiltration of *Nicotiana benthamiana* with the binary vector pCambia-crTMV-CP_PLRV._ The significant levels of the chimeric virus enabled direct visualization of crTMV-CP-PLRV in the cell and to investigate the mechanism of the pathogenesis. Localization of the crTMV-CP-PLRV in plant cells was examined by immunoblot techniques, as well as light, and transmission electron microscopy. The chimera can transfer between vascular and nonvascular tissues. The chimeric virus inoculum is capable to infect *N. benthamiana* mechanically. The distinguishing feature of the chimeric virus, the RNA virus with the positive genome, was found to localize in the nucleolus. We also investigated the role of the N-terminal sequence of the PLRV P3 coat protein in the cellular localization of the virus. We believe that the gene of the PLRV CP can be substituted with genes from other challenging-to-study plant pathogens to produce other useful recombinant viruses.

## 1. Introduction

Potato leafroll virus (PLRV) is the type species of the genus *Polerovirus*, family *Luteoviridae*. It is a highly pathogenic, phloem-limited cytoplasmic virus that accumulates in infected plants in low quantities. The PLRV genome encodes two coat proteins (CP)—the major protein P3 (22.3 kDa) and the minor protein P5 (55 kDa), which is an extended version of the P3 protein produced by translational read-through [1]. PLRV, like other luteoviruses, cannot be transmitted mechanically. It infects the plants only when delivered into phloem tissues by aphids, grafting, or agroinoculation. In infected plants, PLRV is mainly restricted to cells in the vascular system [2,3]. This restriction possibly occurs because PLRV movement functions do not operate in epidermis and mesophyll cells [4] and because PLRV cannot evade silencing-like host-defense responses in nonvascular tissues [5,6,7]. In plants, PLRV is able, to some extent, to exit from the phloem into mesophyll cells, when the PLRV-infected plants are also infected by certain other plant viruses [8,9,10]. The study of infection and virus transport is particularly challenging because the virus is not mechanically transmissible. Moreover, PLRV is a low-titer virus. The yield of the purified virus depends on the host plant but is usually poor. The mean yields of wild-type (WT) PLRV in potato and *Physalis* are 400 and 1300 ng/g leaf [11,12], respectively. Thus, it is challenging and complicated to obtain the virus antigen, the coat protein, to study its role in virus infection and to raise virus-specific antibodies for diagnostics in practice. Therefore, the obtaining of virus antigen, the coat protein, for investigation of its role in viral infection and for increasing of antibodies production for diagnostics, is a challenging and complicated task.

In our previous study [13], we constructed an efficient agroinfiltrable viral binary vector pCambia-crTMV-CP_PLRV_ to overcome the problem of the low PLRV yield; its complete nucleotide sequence (14410 bp) is presented [14]. The vector contains a recombinant cDNA of the crucifer-infecting tobacco mosaic virus (crTMV) [15], in which the coat protein gene replaced by the corresponding PLRV gene. Agroinfiltration of *N. benthamiana* with this binary vector resulted in the multiplication of the tobamovirus RNA and PLRV CP, and the formation of spherical chimeric virus particles tentatively called a chimeric virus, crTMV-CP-PLRV. The morphology, size, and antigenic specificity of the wild-type PLRV and the purified chimeric virus were found to be similar. The yield of crTMV-CP-PLRV was about a thousand times higher than the WT PLRV [11,12]. The virus described in this study has several features in common with tobamoviruses and luteoviruses. P3 coat protein is the only polerovirus component of the chimera; all other parts of it belong to the cytoplasmic tobamovirus. Molecular mechanisms of the chimeric virus import and export in and out of the cell, and its cellular localization are not known. A significant challenge is that different virus families use different strategies for long-distance transport [16]. We believe that after agroinfiltration of *N. benthamiana*, the chimeric virus genome is first transiently expressed in the mesophyll tissue through RNA synthesis in all transfected cells. As with the transport of similar viruses in the infected cell [17], we hypothesize that the chimeric virus RNA translocates to the cytoplasmic membranes where the viral genome is translated by ribosomes and replicated in the replication factories [18,19]. We presume that the transported forms of the virus RNA can be the complex with the movement protein (RNA-MP) or with PLRV CP (vRNP), or the vRNP-MP composite, and even virion [17]. The high levels of the chimera enabled a direct visual study of the chimeric virus in the infected cell. Because of this, we could analyze the localization of crTMV-CP-PLRV, using uninfected plants as a control. Localization of the chimeric virus in plant cells was examined by immunoblot techniques, as well as light and transmission electron microscopies.

## 2. Materials and Methods

Dr. Yu. Varitzev (Potato Farming Institute of the Russian Agricultural Academy) kindly provided tube potato plants infected with wild type PLRV (Canadian strain). Antiserum to chimeric PLRV obtained as described [13]. Antibodies against PLRV and the CMV (cucumber mosaic virus) purchased from DSMZ plant virus collection.

Binary vectors—The crTMV-based vector, pCambia-crTMV-CP_PLRV_, which produced hybrid virus particles, crTMV-CP-PLRV, of chimeric crTMV RNA encapsidated by PLRV CP, was engineered by replacing the crTMV CP gene with the PLRV 22.3 kDa CP (ORF 3) gene [13].

Three control vectors were constructed similar to pCambia-crTMV-CP_PLRV_, where the PLRV CP gene was substituted with three different genes coding GFP and its two modifications (will be published elsewhere). 

Plant growth—*N. benthamiana* plants were grown in a chamber with a cycle of 16 h light at 20 °C and 8 h dark at 15 °C.

Agroinfiltration—*Agrobacterium tumefaciens* strain GV-3101 was chemically (20 mM CaCl_2_) transformed [20] with pCambia-crTMV-CP_PLRV_. Bacteria were grown in 2YT medium containing antibiotics: rifampicin, gentamycin, and kanamycin at 50, 25, and 100 µg/mL, respectively, at 28 °C overnight. The cells were collected and re-suspended in agroinfiltration buffer (10 mM MgCl_2_, 10 mM MES, pH 6.5) to a density of 0.6 at OD_600_. Approximately 200 µl of the transformed bacterial suspension was infiltrated into the abaxial side of *N. benthamiana* leaves, using a syringe without a needle, mostly using 10 plants at 5–6 leaf stage. The injected mixture also contained agrobacterium harboring a plasmid expressing the tomato bushy stunt virus p19, a potent protein-suppressor of RNA silencing [5]. 

Mechanical transmission—First, fresh leaf tissue homogenate of an agroinoculated plant (6–7 days post agroinfiltration) was obtained. For this purpose, leaves of 5–6 leaf stage *N. benthamiana* were mixed with the three-times the weight of 10 mM Tris-HCl (pH 8.0) or 10 mM phosphate buffer (pH 7.5) in a chilled mortar and were homogenized with a pestle with Celite^®^ R 281 (Sigma-Aldrich, Merck, Darmstadt, Germany). Ten uninfected young, at the 5–6 leaf stage, plants with very thin and soft cuticles were used for mechanical transfection with the sap obtained above. Celite powder and 20–30 µL of homogenized sap from the infected plant were rubbed into 1–2 leaves of each plant. As rubbed areas on leaves often become dry, we prevented covering the plant with polyethylene bags to allow the recovery of the cuticle. Symptoms of infections appeared 6 days after inoculation. In parallel, plants were inoculated in the same manner with 20 µg of the chimeric virus previously purified and stored at −20 °C. As a control, WT PLRV was rubbed into 10 young *Nicotiana* plants.

We used quantitative DAS-ELISA to compare the mechanically infected and uninfected plants. For this purpose, 5 mm diameter disks from three leaves of two sample plants with a total mass of approximately 45 µg were obtained with the hole-punch. These discs were ground in 5 µl of PBS-0.05% Tween20 buffer with 1 mL Potter homogenizer. Then, the homogenate was diluted 20-fold with the same buffer and clarified by centrifugation at 1000 rpm for 5 min. Serial double dilutions of the supernatant and the isolated crTMV-CP-PLRV (1 µg/mL) were transferred into the MaxiSorp immune plates (Nunc) covered with antibodies against the chimeric virus and incubated overnight at 4 °C. After washing, the of antibody-virus complex was analyzed with the secondary antibodies conjugated with the horseradish peroxidase as described [21].

Working with the artificial chimeric virus, we were guided by the general rule that infectious material should be sterilized, and the infection should be killed. We ensured that no virus infection would to go outdoors from the lab.

Leaf tissue blot-immunoassay—The epidermis of leaves of *N. benthamiana* were stripped with forceps and scraped off partially with a small brush. These leaves were then pressed onto Protran BA 85 0.45 µm nitrocellulose membrane filters (GE Healthcare, Chicago, MI, USA) under a 400 g weight for 15–20 s. Then, the leaf blots were placed in Petri dishes and incubated for one hour at room temperature with TBS-T buffer (100 mM Tris-HCl, pH 8.5, 150 mM NaCI, 0.1% Tween-20) containing 5% membrane blocking agent (Thermo Fisher Scientific, Waltham, MA, USA). Membranes were subsequently washed three times for 10 min at room temperature in TBS-T and incubated overnight at 4 °C with antiserum to the chimeric virus diluted 1:15000 with TBS buffer (TBS-T without Tween). Then the membranes were washed three times for 10 min with TBS-T and incubated for 1 h at room temperature with goat anti-rabbit secondary antibody conjugated to alkaline phosphatase (Thermo Fisher Scientific, Waltham, MA, USA). Detection performed using a chromogenic substrate of alkaline phosphatase 5-Bromo-4-chloro-3-indolyl phosphate and tetranitroblue tetrazolium (BCIP/NBT) [21]. 

Petiole cross-sections immunoblot analysis— Several prints of N. benthamiana petiole cross-cuts agroinoculated with the chimeric virus or wild type CMV (cucumber mosaic virus), as a control virus that causes a systemic infection, were obtained on 0.45µm nitrate cellulose membranes. Membranes were treated with the blocking agent, incubated with commercial virus-specific antibodies, followed by secondary antibodies, and then analyzed with the chromogenic substrate substantially as described above.

Sample preparations for microscopy— Six days after agroinfiltration, small leaves of the fourth and fifth tier of healthy and infected *N. benthamiana* plants were fixed with 2.5% glutaraldehyde solution (Merck, Darmstadt, Germany) in 0.1 M Sorensen’s phosphate buffer (pH 7.2) and 1.5% sucrose. After washing out the residual glutaraldehyde, the plant material was stained with 1% osmium tetroxide (Merck, Darmstadt, Germany), and dehydrated in ethanol through stepwise increasing concentrations of 30, 50, 70, 96, and 100%. Residual ethanol was then removed with propylene oxide (Chimmed, Moscow, Russia), and the leaf tissue was embedded in a mixture of epoxy resin and Epon 812 Araldite (Merck, Darmstadt, Germany) according to standard procedures [22]. 

For analysis by light-optical microscopy, semi-thin (2.1 µm) tissue sections mounted on glass slides were stained with a 0.1% aqueous solution of methylene blue (Merck, Darmstadt, Germany) and embedded in epoxy resin. Slides were analyzed with the Olympus BX51 microscope (Olympus, Tokyo, Japan) and photographed with a Color View II camera (Soft Imaging System, Munster, Germany). Ultrathin sections were prepared using a diamond knife on an LKB-V ultramicrotome (LabX, Midland, Canada). They then stained with 5% aqueous uranyl acetate and Reynolds lead citrate [23], and visualized using a Jeol 1400 transmission electron microscope.

Genetic engineering of the nucleolar localization sequence (NoLS)—The amino-terminus of PLRV CP has the sequence: ^1^MSTVVVKGNVNGGAHQPRRRRRQSLRRRANRVQPVVMVTAPGQPRRRRRRRGGNRRSRRTGVPRGRGSSE^70^ (Canadian strain, GenBank: D13954.1). Regarding the bipartite nucleus/nucleolus PLRV CP signal, one can conclude that the second sequence, ^45^RRRRRRRGGNRRSRR^59^, is a stronger signal for nucleolar localization than the first sequence, ^18^RRRRRQSLRRR^28^. We deleted 15 nucleotides in the CP gene of pCambia-crTMV-CP_PLRV_, encoding five arginine residues (aa 18–22), to produce pCambia-crTMV-CP_PLRV_(Δ5R) using overlapping PCR with two pairs of primers: 1)5‘-ttctcgagatgagtacggtcgtggttaaaggaaacgtcaacgg-3‘ PLRV-CP (+)2)5‘-ttggtaccctatttggggttttgcaaagcca-3‘PLRV-CP (-)3)5‘-ggtggtgcacaccaaccacaatcccttcgcaggcgc-3‘ PLRV (+)4)5‘-gcgcctgcgaagggattgtggttggtgtgcaccacc-3‘ PLRV (-),

Where (+, -) represent coding and complementary nucleotide sequences. 

## 3. Results and Discussion

### 3.1. Previous TMV/PLRV Synthetic Vectors

Earlier [7], a vector was constructed on the base of the TMV U1, PLRV (TMV∆CP-PLRV-CP), in which the CP PLRV gene replaced the normal CP gene. When the chimeric virus TMV∆CP-PLRV-CP was agroinoculated into *N. benthamiana*, this vector showed limited accumulation in *Nicotiana*. It did not move systemically, likely because the TMV CP is essential for TMV long-distance movement, and PLRV CP was unable to complement this function as was hypothesized. Additionally, Barker et al. [7] examined PLRV infection in *N*. *benthamiana* transformed with *Agrobacterium tumefaciens* containing the plasmid pBNUP110 or in the plant infected by rubbing the PLRV transcripts onto carborundum-dusted leaves. Again, the PLRV yield was no more than 1,600 ng/g of the transgenic plant, even when mixed with PVY infection.

There are several differences between these constructions and those made by us, including different strains of TMV, deletion of regulatory sequences, absence of the P5 protein, and different movement proteins. Additionally, we employed a more efficient method of plant infection using agroinfiltration with the powerful P19 silencer [13], instead of P1/HC-Pro [7].

To characterize interaction between the chimera obtained by us and *N. benthamiana*, we examined the dynamics of the systemic infection spread in the host plant and the chimeric virus localization.

### 3.2. Chimeric Virus Infectiousness

Agroinfiltration of the chimeric virus caused a severe infection of *N. benthamiana*. After two-to-three days following infiltration with pCambia-crTMV-CP_PLRV_, the inoculated region of *N. benthamiana* became necrotic. Moreover, six days post-agroinfiltration, petioles of the upper tier of leaves became much thinner in addition to showing necrosis. The leaves associated with these petioles were shrunken and were smaller in size (Figure 1a,b). The leaf tissue blot-immunoassay confirmed the chimeric virus infection of plants. Furthermore, upper leaf tissue blot-immunoassay confirmed visible symptoms of severe systemic infection of *N. benthamiana* with the chimeric virus. The most intense staining of infected *N. benthamiana* was in the vascular tissue of the leaves (Figure 1c). 

As a control, we constructed three binary vectors pCambia-crTMV-CP_PLRV_, in which gene of the PLRV CP was substituted with the gene of the GFP to produce pCambia-crTMV-GFP_α,β,γ_ (Figure 2). After 7 days of agroinfiltration of *N. benthamiana* with these vectors, no necrosis of petioles and leaves was observed. Figure 2b illustrates that infection with these vectors had local character and developed slowly. The first indications of infection appeared at the upper leaves after four weeks of agroinfiltration. From this, we inferred that the coat protein of PLRV is essential for the systemic infection of *N. benthamiana* by the chimeric virus.

### 3.3. crTMV-CP-PLRV Exits the Phloem Into Nonvascular Tissues of N. benthamiana

We analyzed whether the severe systemic infection and high virus productivity of the chimera was due to it exiting the vascular bundle or whether the virus remained confined to the phloem tissue, as luteoviruses do [24]. As a control for comparison, we chose the cucumber mosaic virus (CMV) which infects plant cells systemically [25]. 

Localization of crTMV-CP-PLRV was examined for 5–7 days post-inoculation of *N. benthamiana* by immunoblot of petiole cross-cuts onto membrane filters followed by light microscopy. Immunoprints of cross-sections of *N. benthamiana* petioles showed that the chimeric virus exited the phloem and spread into non-vascular tissues (Figure 3b) like the cucumber mosaic virus (Figure 3a). 

### 3.4. Fresh Virus Inoculum Can Infect Mechanically

We also found that the inoculum of the chimeric virus, but not WT PLRV, can infect mechanically, causing a very similar infection for all ten *Nicotiana* plants in comparison with agroinfiltration (Figure 1b). This observation was confirmed by DAS-ELISA of the crTMV-CP-PLRV infected and uninfected *N. benthamiana* (see Figure 4). Our calculations showed that the yield of the mechanically infected crTMV-CP-PLRV was comparable (about 0.4 mg per g of the plant) to the yield of the virus obtained by pCambia-crTMV-CP_PLRV_ agroinfiltration (about 0.8 mg per g of the plant).

Both of these findings were consistent with Peter et al. 2009 [26], who showed that mutated PLRV that lacked the read-through C-terminal portion of the P5 coat protein exited phloem to the mesophyll, and whose progeny were temporarily able to infect *Solanum sarrachoides* mechanically. 

Surprisingly, fresh virus inoculum in plant sap but not purified virus preparations that were kept frozen at −20 °C was infectious mechanically. The likely explanation is the virus RNA degradation before or during its extraction from the frozen chimeric virus [13]. Bald [27] observed the digestion of RNA from tomato spotted wilt virus unless the whole virus genome is covered by a protein coat. We conclude that the chimeric virus capsid does not protect RNA completely from degradation by nucleases, particularly after freezing. There are two possible explanations: first, that the virus capsid is imperfect because the chimeric RNA is 640 nucleotides longer than WT PLRV RNA, and icosahedral viruses are limited in the amount of RNA they can carry. Second, this may be due to the absence of the PLRV read-through P5 protein. Thus, it appears that the chimeric RNA is in some way accessible to nucleases (and potentially importins) inside the capsid, due to the presence of defects or gaps. 

### 3.5. Chimeric Virus Infection Induces Necrosis

In this study, 5 non-agroinfiltrated plants and 10 agroinfiltrated plants were used in the experiment. Under a low magnification of light microscopy (250×), internal and external phloem cells, companion cells of sieve tubes, xylem, and parenchyma were visible at the base of petioles (Figure 5a). *N. benthamiana,* a representative of the family *Solanaceae*, has internal and external phloem tissues and large parenchyma cells with large vacuoles, surrounded both xylem and phloem. Companion cells in the leaf stalks are much larger than sieve elements. Their characteristic features are large nuclei and dense cytoplasm because these cells usually have a significantly larger number of ribosomes and mitochondria. In sieve tubes, the cytoplasm is more often seen near the thickened walls as a thin layer. 

Because of an unusually large number of chimeric virus particles in the infected plant compared to WT PLRV infection, it was easy to determine their cellular localization, using uninfected plants as a control [28,29]. To test this, fixed semi-thin sections of petioles and veins of fourth tier leaves of *N. benthamiana* from uninfected (Figure 5a) and infected plants (Figure 5b), five-to-seven days post-inoculation, were examined under light microscopy at low magnification. We observed virus in the primary phloem cells of the vascular bundle and companion cells of the infected plants (Figure 5b,c) but not in the uninoculated plants (Figure 5a). Internal and external phloem necrosis and necrosis of adjacent cells had occurred in the vascular bundle and companion cells. Areas of necrosis reaching significantly into adjacent cells, including the cells of the parenchyma, were visible (Figure 5b,c). 

As was earlier observed [13], even at an early stage of *N. benthamiana* infection by the chimeric virus, there was a substantial degradation of host and chimeric virus RNAs, as well as necrosis of the leaf petiole. This observation led us to conclude that the infection with crTMV-CP-PLRV induced profound hydrolysis of cellular and viral RNAs during necrosis. Using an experimental system of potato virus X-based recombinant, PVX-Vp26, earlier Wieczorek et al. [30] have previously demonstrated that Vp26, the coat protein of tomato torrado virus, was an effector protein of a lethal systemic necrosis phenotype in plants where Vp26 was expressed in the context of a viral infection. 

### 3.6. Intracellular Localization of crTMV-CP-PLRV

The majority of RNA viruses are replicated in the cytoplasm of infected cells, including transcription, replication of the RNA genome, and assembly of new infectious particles. Typical of tobamoviruses, crTMV, as well as its homolog TVCV, are cytoplasmically localized. On the other hand, Shepardson et al. [31] have observed many wild type PLRV particles in the cytoplasm and other compartments of infected potato, with a few in the nucleus. 

According to Haupt et al. 2005 [32], during challenge inoculation with TMV of transgenic *N. benthamiana* plants with a firmly silenced full-length PLRV genome, spherical PLRV virus particles were found confined to the cytoplasm of infected cells. However, PLRV CP was also located in the nucleolus by gold-labeled polyclonal antibodies to PLRV. Several studies described the localization of some viruses [31,33,34] in the nucleus with some viral proteins being found in the nucleolus [32,35,36,37,38]. 

Earlier, we showed that the crTMV-CP-PLRV is infectious by agroinfiltration and, in this work, mechanically, that it is a chimeric virus, not VLP (virus-like particles) [13]. Because of the cytoplasmic location of crTMV and cytoplasmic and nuclear location for the luteovirus PLRV coat protein [32], we examined whether the crTMV-CP-PLRV was confined in the cytoplasm or the nucleus. 

In this study, three non-agroinfiltrated plants and six agroinfiltrated plants were examined independently, while the earlier-characterized chimeric virus was isolated from the same infected plants. High-resolution transmission electron microscopy (TEM) was used to obtain images of fixed, ultra-thin sections of petioles and veins of fourth-tier leaves of uninfected and infected *N. benthamiana* plants. Infection with crTMV-CP-PLRV dramatically changed phloem cells, sharply reducing the number of cells containing a nucleolus and decreasing the content of chromatin in the nucleus. In the early stages of infection, at 5–7 days post-inoculation, numerous viral particles appeared in the majority of leaf blade cells, i.e., in infected companion cells, cells of phloem, and xylem, and parenchyma cells adjacent to the vascular bundle. Virus particles formed dense clusters, localized predominantly in the nucleolus, nucleus, and at a much lower level in the chloroplasts and vacuoles. Cells of the vascular bundle containing nuclei, i.e., primary phloem cells and companion cells from uninfected and infected plants 6–7 days post-inoculation, were selected for more detailed study. Images of more than 2000 cells of infected plants (341 cells contained nuclei), and more than 500 for control samples of uninfected cells (100 cells contained nuclei), were compared. Figure 6a–e show serial cell images of plants that were assembled to demonstrate images with the enhanced resolution. 

High-resolution TEM study at the sub-cellular level of crTMV-CP-PLRV infection showed that the virus first appeared in the granular component of the nucleolus (Figure 7a,b). This finding was consistent with both the crTMV MP and PLRV CP P3 containing signals for nuclear localization [15,32].

The chimeric virus continued to accumulate in the nucleolus, displacing the contents of this organelle. The virus then spread out of the nucleolus and filled the entire volume of the nucleus, where it formed multilayered honeycomb-like pseudo-crystal structures (Figure 8a). By this stage, chromatin had mostly disappeared, and the virus exits the nucleus into the cytoplasm after the nuclear membrane lysis (Figure 8b). As a result, necrotic cells were observed in infected areas, along with apparently intact cells. 

In rare instances, we also observed also clusters of the chimera virus particles in chloroplasts (Figure 9a,b).

### 3.7. Major PLRV Coat Protein Targets the Chimeric Virus to Nucleolus

The TMV U1 MP (P30) lacks nuclear localization signal (NLS) sequences, which are present in the crTMV MP [15] and which are conserved in the *Tobamoviruses* of subgroup III, but not in *Tobamoviruses* of subgroups I and II, including TMV U1 [39,40]. Earlier, NLS sequences were mutated in the gene encoding P30 MP of turnip vein clearing virus (TVCV), which belongs to the same subgroup as crTMV. In addition to blocking the nuclear localization of P30 [41], this resulted in the suppression and delay of TVCV systemic transport in *N. benthamiana*, suggesting that the nuclear function of P30 is needed for active systemic infection, but not for the nucleus localization.

The coat protein of PLRV contains in its structure both NLS and NoLS signals [32]. Short sequences of 6–10 basic amino acids (aa) called NLSs, some of which also serve as NoLSs, target the nuclear import of many proteins. Recent systematic analysis of 46 NoLSs for their amino acid composition and sequence features has shown greater sequence diversity, including bipartite nucleus/nucleolus sequences, as well as overlap with NLSs [42]. Salvetti and Greco [37] have reported that the import of some virus capsid proteins into the nucleolus is rapid and varies according to the nucleolar localization sequences. In addition, increasingly, data shows that plant viruses hijack the nucleolus to promote virus replication [35]. 

Interestingly, agroinfiltration of five *N. benthamiana* plants with pCambia-crTMV-CP_PLRV_(Δ5R) resulted in an active and severe infection very similar to the infection of pCambia-crTMV-CP_PLRV_, precisely as shown in Figure 1b. Therefore, the deletion of the five arginines, aa 18–22, did not change the target properties of PLRV CP in the chimeric virus. Thus, we conclude that sequence ^45^RRRRRRRGGNRRSRR^59^ targets the coat protein and the virus to the nucleolus. 

### 3.8. Transport of the Chimeric Virus Infection

The chimeric virus has features of both viruses. As tobamoviruses, the chimera grows to high concentration. This virus transfers between vascular and nonvascular tissues, and is infectious mechanically. As poleroviruses, it is spherical and is recognized by antibodies to PLRV [13], its capsid protein has signal amino acid NLS and NLoS sequences. As in the case of coat proteins of other viruses [16,17,38], the plant cell nucleus is targeted by the PLRV CP. We suggest that the PLRV P3 protein is the movement factor that is responsible for trafficking virus RNA-protein complex to the nucleolus where matured virus particles probably assembled [16].

It should be emphasized that all other parts of the chimeric virus belong to the cytoplasmic tobamovirus, with the PLRV P3 coat protein being the only polerovirus component of the chimera. It appears the PLRV CP has to be a multifunctional protein as many other virus coat proteins involved in the particle assembly [43] and long-distance viruses transport within host plants [44,45,46]. As known, in addition to cell-to-cell and long-distance transport within plants, viral capsid proteins play a significant role in the infection process, including symptom development and pathogenicity [47,48]. 

We propose that the PLRV CP binds to the tobamovirus RNA, thus protecting the transfer form of the virus for moving through phloem tissue and substitutes the tobamoviral CP in systemic transport.

## 4. Conclusions 

Summarizing the findings from obtained data, we can state that the CP of PLRV can substitute for the tobamoviral CP to produce powerful virus. The major coat protein of the potato leafroll virus targets the chimeric virus with the positive RNA genome to the nucleolus of *Nicotiana benthamiana* cells. The fresh chimeric virus inoculum can mechanically transmit to *N. benthamiana*. The chimeric virus exits phloem tissue.

The chimera is a robust system that gives high CP yields thus overcoming a serious limitation in luteovirid research. We believe that the gene of the PLRV CP can be substituted with genes of other difficult-to-access plant pathogens to produce other useful recombinant viruses.

The particles of the obtained chimeric virus may be a fascinating object of study under cryo-electron microscopy [49] taking into consideration that the structure of the ryegrass mottle virus (RMV) capsid was resolved at 2.9 angstroms [50] and that most non-enveloped icosahedral viruses have the same type of icosahedral capsids.

## Figures and Tables

**Figure 1 high-throughput-09-00011-f001:**
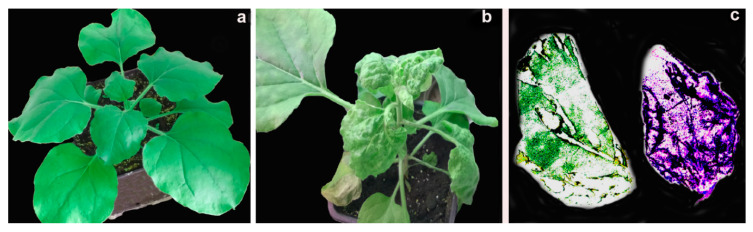
pCambia-crTMV-CP_PLRV_ induces severe infection of *N. benthamiana.* Leaves of the fourth tier (from the top) of infected plants were examined six days after agroinfiltration of *N. benthamiana* with pCambia-crTMV-CP_PLRV_**.** (**a**) non-inoculated plant; (**b**) *N. benthamiana* 6 days post-agroinfiltration with pCambia-crTMV-CP_PLRV_; (**c**) leaf tissue blot-immunoassay for detection of virus infection. Left—healthy; Right—crTMV-CP-PLRV infected *N. benthamiana*.

**Figure 2 high-throughput-09-00011-f002:**
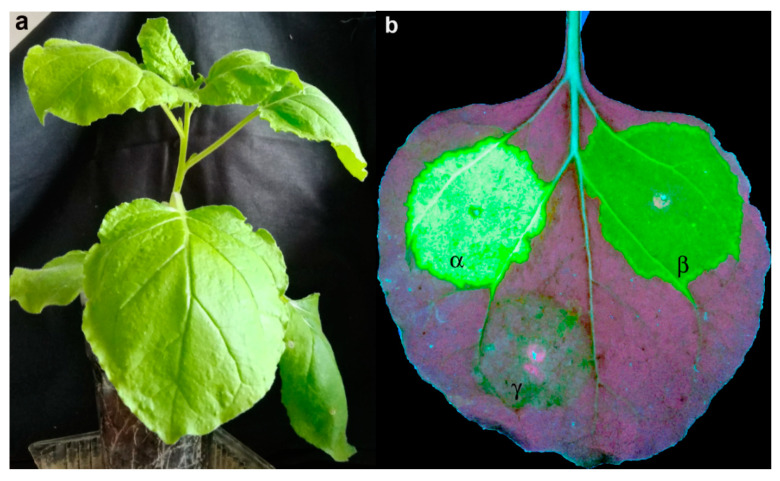
Replacing the PLRV CP gene for the GFP gene resulted in slower rates of infection. (**a**) 7 days after agroinfiltration, the plant looks quite healthy; (**b**) α, β, γ show areas of agroinfiltration with pCambia-crTMV-GFP_α,β,γ_ vectors. GFP fluorescence was observed at 354 nm.

**Figure 3 high-throughput-09-00011-f003:**
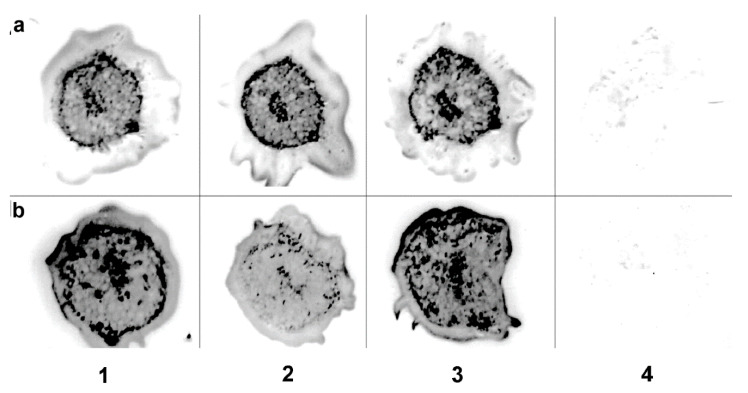
Chimeric virus exits the phloem into nonvascular tissues of *N. benthamiana*. Immunoblots of cross-sections of *N. benthamiana* petioles were performed after tissue-printing onto 0.45 µm nitrate cellulose membrane. Blots were developed with commercial antibodies against CMV and WT PLRV and conjugates of secondary antibodies with alkaline phosphatase. Images were obtained on an Epson scanner with 3200 dpi resolution. Images in rows 1, 2, and 3 show blots of petioles crosscuts of the different plants. (**a**) Typical picture of the WT CMV infection, row numbers 1–3. The distribution of the WT virus is quite homogenous with some prevalence in the conducting bundle. (**b**) the chimeric (virus rows 1–3) can also be found in most cells of petiole together with small spots of high virus antigen concentration (possibly due to necrosis). Row 4 is the negative control of immunoblots. No specific labeling of cells was observed in healthy control tissue used in these experiments (see row 4 in Figure 3a,b).

**Figure 4 high-throughput-09-00011-f004:**
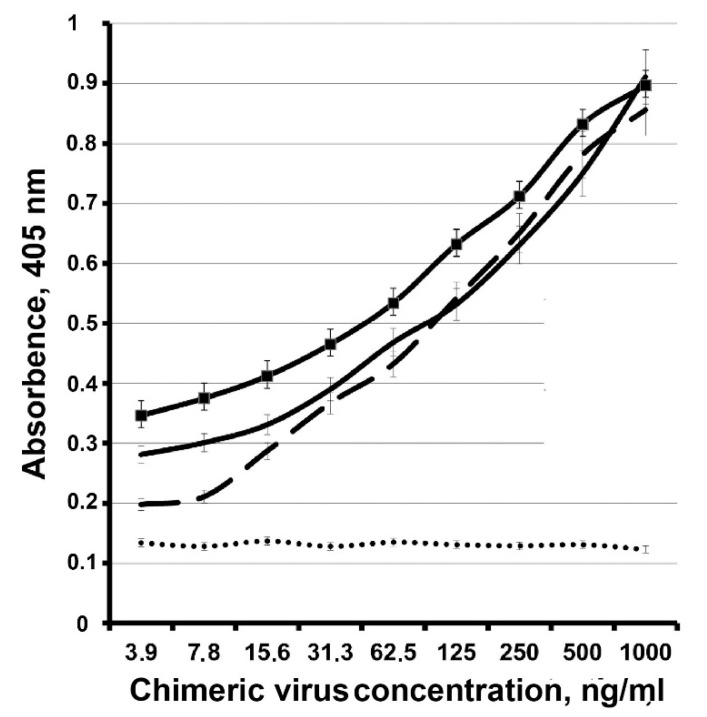
DAS-ELISA of the mechanically-infected *N. benthamiana*. Chimeric virus preparation was a positive control, whose quantity was estimated by UV-absorption spectroscopy [13]. Uninfected *N. benthamiana* was used as a negative control. The standard deviation measured the dispersion of data. Line designations: dotted—uninfected plant, negative control; dashed—a known quantity of the chimeric virus added to the uninfected plant extract sample, positive control; solid lines—samples 1 and 2 of the chimeric virus mechanically-infected *N. benthamiana*.

**Figure 5 high-throughput-09-00011-f005:**
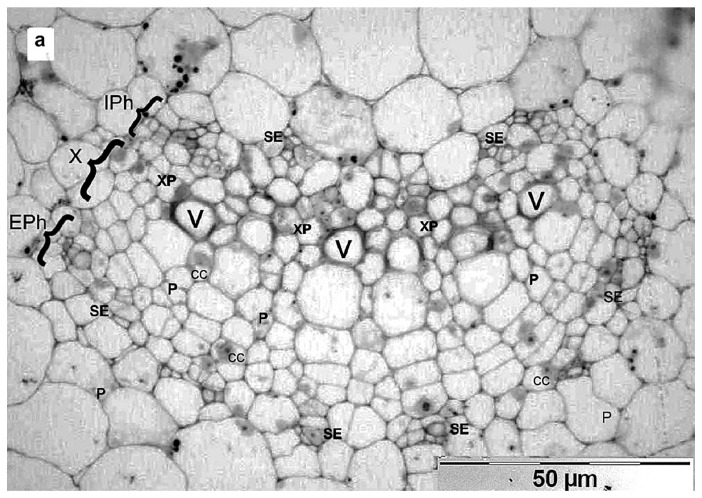
Necrosis of phloem and adjacent cells in *N. benthamiana* at an early stage of infection. Light microscopy study. (**a**) Image of healthy plant; (**b**) image of agro-inoculated *N. benthamiana* petiole base; (**c**) image of an infected *N. benthamiana* leaf. Red arrowheads indicate necrosis location. EPh—external phloem, IPh—internal phloem, CC—companion cell, SE—sieve element, V—vessel, XV—xylem vessel, XP—xylem parenchyma, X—xylem, P—parenchyma.

**Figure 6 high-throughput-09-00011-f006:**
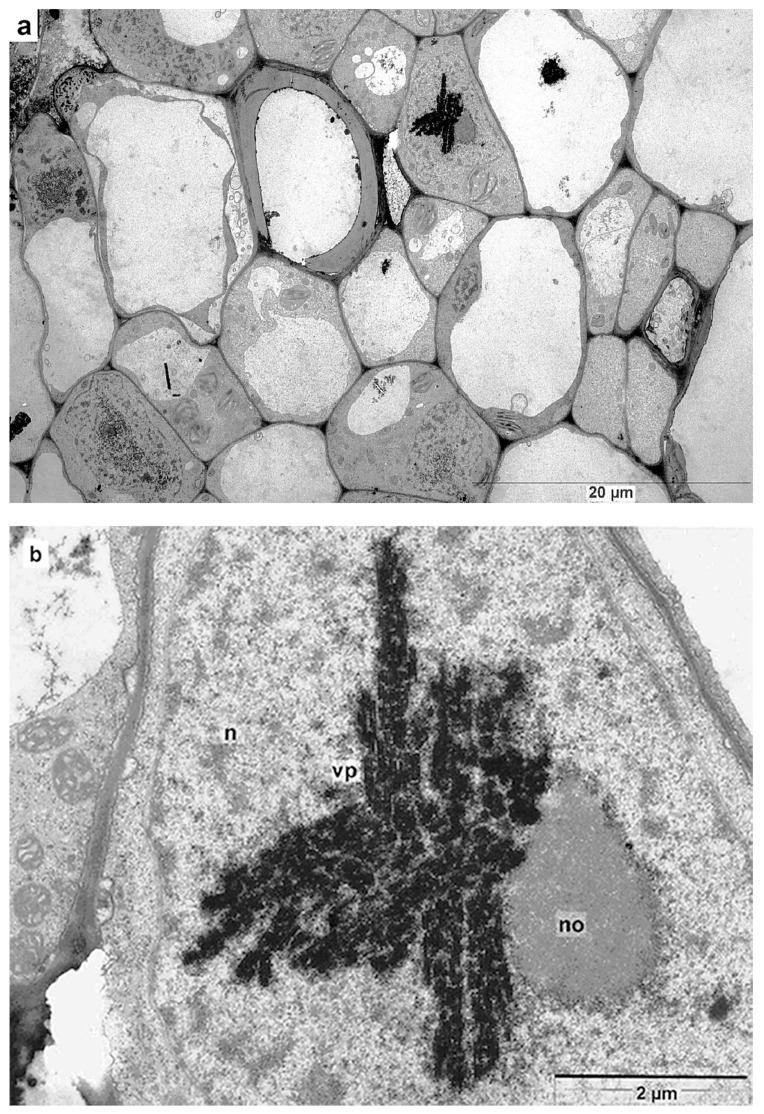
Infected with crTMV-CP-PLRV and uninfected phloem cells. **a**,**b**,**c** and **d**,**e** (uninfected, negative control) images are taken at increasing magnification. Compare infected and uninfected cells in images (**b**) versus (**d**) and (**c**) versus (**e**) taken under similar magnification. ‘no’ indicates the nucleolus, ‘ch’ indicates chromatin.

**Figure 7 high-throughput-09-00011-f007:**
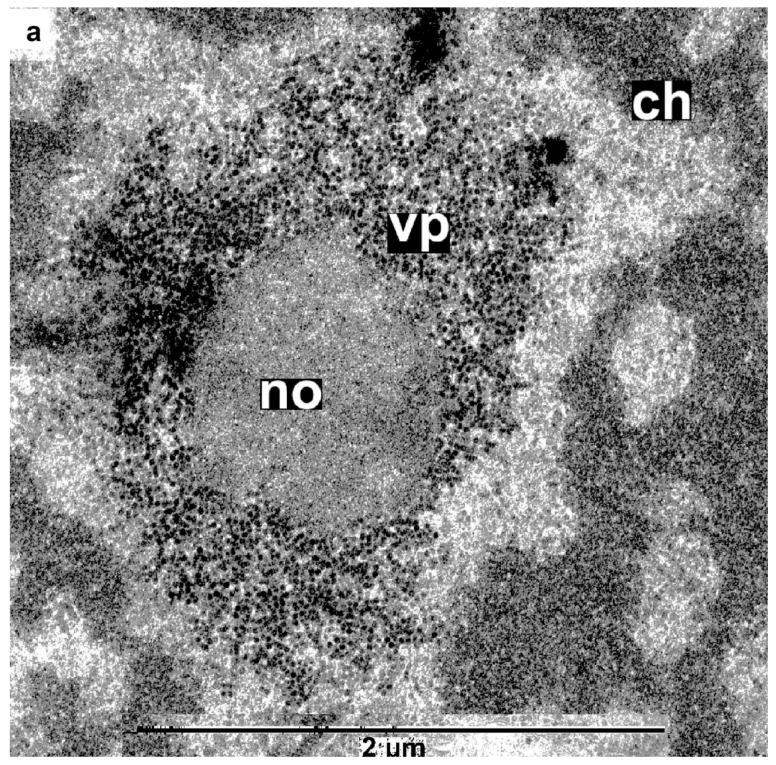
Initial appearance of chPLRV in nucleolus. (**a**,**b**) the virus occupies the granular layer of the nucleolus.

**Figure 8 high-throughput-09-00011-f008:**
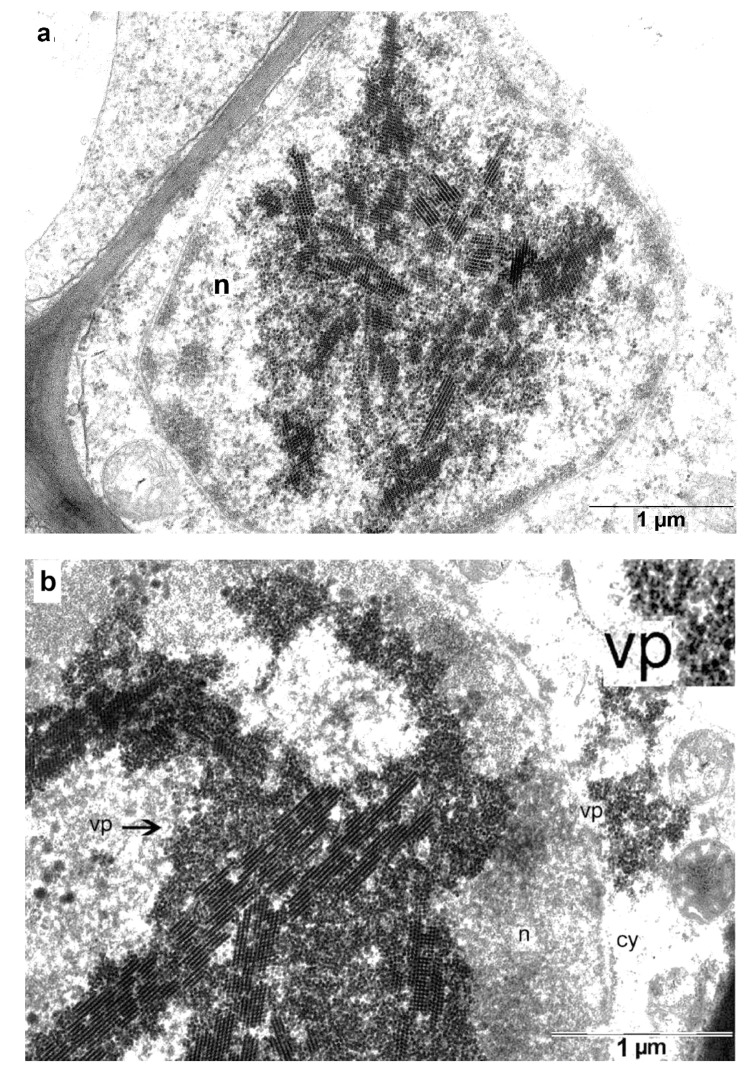
Development of the chimeric virus infection in the infected cell. (**a**) Invasion of the nucleus by the chimeric virus; (**b**) The chimeric virus spreads throughout the entire volume of the nucleus and moves into the cytoplasm of the infected cell through a gap in the nuclear membrane. Insert in the right upper corner is a tri-fold magnified image of the virus particles outside of the nucleus. n—nucleus, no—nucleolus, vp—virus particles, ch—chromatin, cy—cytoplasm.

**Figure 9 high-throughput-09-00011-f009:**
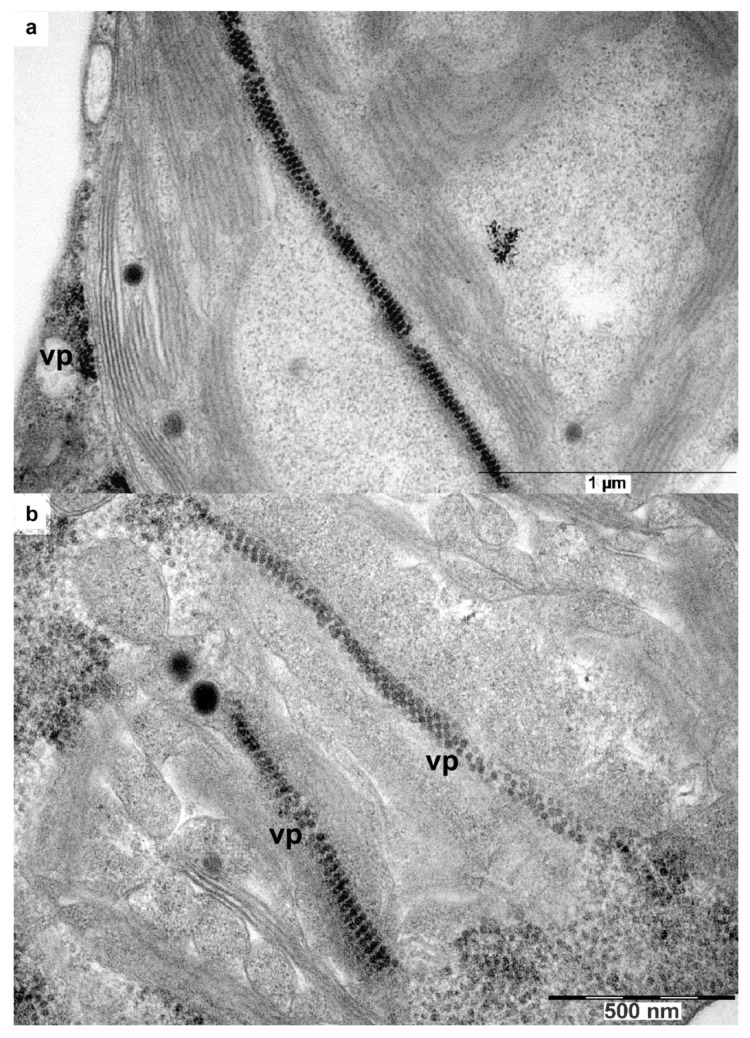
Chimeric virus localization in chloroplasts. (**a**,**b**) intracellular localization of the chimeric virus in phloem cells (core parenchyma), with clusters of the chimera virus particles arranged along with the chloroplast envelope. VP—virus particles.

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
