# Peer review of "Chimeric Virus Made from crTMV RNA and the Coat Protein of Potato Leafroll Virus is Targeted to the Nucleolus and Can Infect Nicotiana benthamiana Mechanically"

_2571-5135, 2020, doi:10.3390/ht9020011_

Round 1

Reviewer 1 Report

The main finding of this study is the development of a chimeric virus made from crucifer-infecting TMV (crTMV) RNA and the potato leafroll virus (PLRV) coat protein (CP) by agroinfiltration of N.benthamiana.

This study suggests that this chimeric virus crTMV-CP-PLRV can substitute the previously developed (PMID: 28921459) chimeric virus composted by the crucifer-infecting TMV (crTMV) and RNA and the tobamoviral coat protein. This is the first work describing crTMV-CP-PLRV.

In my opinion, this work is relevant to the field. I have only few specific comments and I am thus recommending a minor revision.

METHODS

Page 2, line 91, “Then” -->Then,

Page 3, lines 100-101, the style need to be revised

Page 3, lines 100-113, pleas add references in this section

Page 5, lines 193-194, please remove “see”

Page 5, line 196-197, ”mg per g” and “mg/g”, the style need to be revised

RESULTS/DISCUSSION

In general, the number of figures can be reduced. For instance, figure 1 and 2 can be compacted in one figure, as well as figures 8, 9 and 10.

Page 11, Figure 6 is not quoted in the text

Page 18, line 357, please align the line

Page 19 line 409, [28-31,36,38). -->[28-31,36,38].

Page 19, line 434, this sentence is lacking in references, for instance, the polyomavirus Simian virus 40 is an important example of unenveloped icosahedral virus (PMID:31403031, doi: 10.3389/fonc.2019.00670).

Page 19, line 442, please, align the line.

REFERENCE

In this section, the reference style needs to be highly and accurately revised/uniformed.

Author Response

Responses (R), critical comments (CC)

Dear Reviewer,             

Thank you for your careful review of our manuscript titled “The chimeric virus made from crTMV RNA and the coat protein of potato leafroll virus targeted to the nucleolus and infected N. benthamiana mechanically.”

I think the manuscript has been significantly improved after the recommended corrections.

CC-1.  Page 2, line 91, “Then” -->Then,

R1.                  Done

CC-2.  Page 3, lines 100-101, the style need to be revised

R2.                  Done

CC-3.  Page 3, lines 100-113, pleas add references in this section

R3.                  Please find reference #21.

CC-4.  Page 5, lines 193-194, please remove “see”

R4.                  Done

CC-5.              Page 5, line 196-197, ”mg per g” and “mg/g”, the style need to be revised

R5.                  Done

CC-6.  In general, the number of figures can be reduced. For instance, figure 1 and 2   can be compacted in one figure, as well as figures 8, 9 and 10.

R6.      Agree, figures  1, 2 and 8, 9 combined.

CC-7.  page 11, Figure 6 is not quoted in the text.

R7.      Please pay attention to page 9, line 290.

CC-8.  Page 18, line 357, please align the line.

R8.                  Done

CC-9.  Page 19 line 409, [28-31,36,38). -->[28-31,36,38].

R9.                  Done

CC-10. Page 19, line 434, this sentence is lacking in references, for instance, the                          polyomavirus Simian virus 40 is an important example of unenveloped                               icosahedral virus (PMID:31403031, doi: 10.3389/fonc.2019.00670).

R10.   I agree, there are the DNA icosahedral viruses. Here the list of the spherical viruses was limited in favor of the RNA viruses.

CC-11. Page 19, line 442, please, align the line.

R11.    Done

CC-12. In this section, the reference style needs to be highly and accurately                                   revised/uniformed.                                                                                                   

R12.    Corrected now. Sorry, I did not succeed in checking out the results of the automatic formatting system.                                  

Reviewer 2 Report

This manuscript described a novel chimeric virus that originated from crucifer infecting TMV and potato leafroll virus. This chimeric virus inherits unique characteristics from both original viruses. It is capable of infecting nonvascular tissues of Nicotiana benthamiana mechanically and expressing the coat protein of potato leafroll virus (P3 protein). During the early stage of infection, this chimeric virus causes necrosis of phloem and adjacent cells. Elctromicroscope images show that this chimeric virus assembles in the nucleus.

The high-quality images provided in this manuscript sufficiently support most of the conclusions, and I would like to add some comment:

Line 42, "In our previous study [13], we constructed an efficient agroinfiltrable viral binary vector...", the citation leads to a general molecular cloning protocol. No information about the viral binary vector pCambia-crTMV-CPPLRV is provided in this citation. I think the citation number might be [10].
Line 61, "...PLRV obtained as described [13]." This citation does not link to any information about the antiserum to chimeric PLRV.
Line 86. "...rubbed into ten yang Nicotiana plants." I think there is a typo.
Figure 1. I think an extra control of N. benthamiana infiltrated with blank Agrobacterium tumefaciens strain GV-3101 and Agrobacterium harboring a plasmid expressing the tomato bushy stunt virus p19 should be provided to eliminate the possibility that Agrobacterium and the agroinfiltration procedure cause necrosis.
Line 158-160. "Figure 2 shows much stronger staining and higher content of the virus antigen (CP PLRV) in agroinoculated N. benthamiana in comparison with the same antigen of wild type PLRV in infected S. tuberosum. " I think it is improper to compare protein expression levels between two plants with this method, especially when Line 167-168 states: "Weak staining of the central vein (Figure 2d) is explained by a light press during blotting of the potato leaf." If light pressing can cause weak staining, it is not convincing that two leaves are pressed exactly the same. Protein purification/Western blotting or another method would support this conclusion better.
Line 220. "Earlier [7], a vector was constructed on the base of the TMV U1, PLRV (TMVΔCP-PLRV-CP)..." Citation 7 did not provide any information about TMV U1, PLRV (TMVΔCP-PLRV-CP).
Figure 5. Images in figure 5 are in different magnification power. 5a is "Image of healthy plant", and 5c is "Image of an uninfected N. benthamiana leaf". It is hard to compare images in different magnification power. And 5a seems not necessary.
Line 262. Citation 13 does not provide any earlier observation.
Figure 6 and 7. Images are in different magnification power. Hard to compare images in different magnification.
Section 3.7. Major PLRV coat protein targets the chimeric virus to nucleolus. I would suggest moving cloning and primer sequences to Materials and Methods.

Author Response

Responses (R), critical comments (CC)

Dear Reviewer,             

Thank you for your careful review of our manuscript titled “The chimeric virus made from crTMV RNA and the coat protein of potato leafroll virus targeted to the nucleolus and infected N. benthamiana mechanically.”

I think the manuscript has been significantly improved after the recommended corrections.

CC-1. Line 42, "In our previous study [13], we constructed an efficient                                           agroinfiltrable viral binary vector...", the citation leads to a general                                       molecular cloning protocol. No information about the viral binary vector                                   pCambia-crTMV-CPPLRV is provided in this citation. I think the citation number might be [10].

R1.      Sorry, I did not succeed in checking out the results of the automatic formatting system. Please find the correct version of the issue REFERENCES.

CC-2. Line 61, "...PLRV obtained as described [13]." This citation does not link to any                              information about the antiserum to chimeric PLRV.

R2.      Please see R1.

CC-3.  Line 86. "...rubbed into ten yang Nicotiana plants." I think there is a typo.

R3.      Corrected.

CC-4.  Figure 1. I think an extra control of N. benthamiana infiltrated with blank                             Agrobacterium tumefaciens strain GV-3101 and Agrobacterium harboring                         a plasmid expressing the tomato bushy stunt virus p19 should be provided                  to eliminate the possibility that Agrobacterium and the agroinfiltration                                   procedure cause necrosis.

R4.      We had other controls. We constructed three binary vectors, in which gene of the PLRV CP was substituted with the gene of the original GFP and its two modifications (will be published elsewhere). We found that agroinoculation nicotiana with these vectors developed locally even after 7 days post-inoculation (Figure 2 added).

CC-5.  Line 158-160. "Figure 2 shows much stronger staining and higher content                           of the virus antigen (CP PLRV) in agroinoculated N. benthamiana in                              comparison with the same antigen of wild type PLRV in infected S.                                             tuberosum. " I think it is improper to compare protein expression levels                                between two plants with this method, especially when Line 167-168                              states: "Weak staining of the central vein (Figure 2d) is explained by a                                light press during blotting of the potato leaf." If light pressing can cause                               weak staining, it is not convincing that two leaves are pressed exactly the                                    same. Protein purification/Western blotting or another method would                                  support this conclusion better.

R5.      I agree with your critical comment. We did not have Nicotiana infected with the WT PLRV. The infection procedure, as you know, needs a natural vector, which is the peach-potato aphid for this virus. The picture with the infected potato excluded.                                                            On the other side, the 400 g press was the same for both Nicotiana and potato leaves. We did compare because the difference in images was proportional to the known data on the yield of the WT PLRV (reference 11, Rowhani et al.) and chimera obtained by us (approximately as 1:1000). 

CC-6.  Line 220. "Earlier [7], a vector was constructed on the base of the TMV                              U1, PLRV (TMVΔCP-PLRV-CP)..." Citation 7 did not provide any                                            information about TMV U1, PLRV (TMVΔCP-PLRV-CP).

R6.      I am sorry. I did not succeed in checking out results of the automatic                                  formatting system after submitting. Please find correct version of the issue                  REFERENCES.

СС-7.  Figure 5. Images in figure 5 are in different magnification power. 5a is                                "Image of healthy plant", and 5c is "Image of an uninfected N.                                     benthamiana leaf". It is hard to compare images in different                                       magnification power. And 5a seems not necessary.

R7.      Thank you for this critical comment. It was my mistake. Fig. 5c is the image of infected                plant. I did correction with magnifications of pictures as well.

СС-8.  Line 262. Citation 13 does not provide any earlier observation.

R8.      Please see R6.

CC-9.  Figure 6 and 7. Images are in different magnification power. Hard to                                   compare images in different magnification.

R9.      I did correction of the pictures magnifications.

CC-10. Section 3.7. Major PLRV coat protein targets the chimeric virus to                          nucleolus. I would suggest moving cloning and primer sequences to                             Materials and Methods.

R10.    I did the correction and transferred part of the section 3.7.to the ‘Materials and             Methods’ section titled “Genetic engineering of the NoLS.”

Reviewer 3 Report

The manuscript by Butenko et al describes the infectivity of the chimeric virus made of crTMV RNA and PLRV coat protein. The chimeric virus can successfully be transmitted mechanically to N. benthamiana. The PLRV CP targets the virus to the nucleolus. The authors also described the intracellular transportation of the virus. The experiments are carefully executed and well described. Overall this is an interesting paper.

Major comments:

The organization of abstract is a bit off. It seems like brief results. Usually an abstract starts with a very brief introduction followed by current research and in the end, just one sentence indicating the importance of work.

English needs to be improved at various places in the manuscript.

Line 151: The authors concluded that the CP is essential for the infectivity, however the data is missing. If authors are concluding anything without the details of the results, it is hard for the readers to understand it well without a proper figure.

Figure 2, why authors used two different species for the WT and chimera virus infection? It is difficult to interpret the results if the two viruses are being infected in two different plants. Because we do not know if the difference is due to the virus or because of the two different plant species.

Figure 3: what is the difference in rows 1, 2 and 3? It should be labelled properly or mentioned in the figure legends/results. Are the rows 1, 2, and 3 same samples at different time points? If yes, then why there is less staining in b-2 as compared to 1 and 3?

Result section 3.4 is a bit complicated. Which vectors were constructed before and which one in this study? Not clearly mentioned. And if it is just the comparison of the chimeric virus in this study and the previous ones, it should be discussed with section 3.1. There is no need to make a separate section for this discussion.

Figure 5: What is the difference in figure 5a and 5c? It is always a good idea to show comparison at the same scale in image so that the difference can be seen clearly. 5b is represented as an image of agro-inoculated N. benthamiana petiole base. Can the authors provide the similar image at similar scale with non-infected N. benthamiana petiole base? It would improve the quality of results.

Figures 5b and 6b look exactly same. However, they have been described as uninfected and infected respectively. Is there some mistake? Similarly, figures 5a and 6a are exactly same, though they were taken at different magnifications.

Section 3.7 describes the presence of NLS and NoLS signals in PLRV CP which could be the reason for the localization of chimeric virus to nucleus. All these studies were done before as referenced by the authors, so there is no point in discussing these again here. Authors can either remove this section or describe very briefly in section 3.6.

Likewise, section 3.8 is more kind of discussion, not results. So instead of discussing it in a separate section, I would recommend to include the content of this section in the corresponding result section. Also, the sentence arrangement in this section is not thorough. It looks more like repetition of the sections described before in the manuscript.

Minor problems:

Line 30: luteoviruses has been introduced first time. It needs a bit more explanation.

We already know that the chimeric virus is infectious, so what do authors want to show new by the results in section 3.1.

Line 221: what is normal CP?

Line 238: the introduction of N. benthamiana should be done earlier, not in the result section.

References are not arranged properly.

Line 451: “We cannot also exclude the possibility that the gene of the PLRV CP can be fused with the polyribonucleotides coding epitopes of pathogenic bacterial, plant, animal, or human proteins to produce vaccines.” This sentence is very broad and cannot be written without proper explanation, references and discussions. It would be best for the manuscript if this conclusion is more focused.

Author Response

Responses (R), critical comments (CC)

Dear Reviewer,             

Thank you for your careful review of our manuscript titled “The chimeric virus made from crTMV RNA and the coat protein of potato leafroll virus targeted to the nucleolus and infected N. benthamiana mechanically.”

I think the manuscript has been significantly improved after the recommended corrections.

CC-1.  The organization of abstract is a bit off. It seems like brief results.                                                   Usually an abstract starts with a very brief introduction followed by                                                 current research and in the end, just one sentence indicating the                                                     importance of work.

R1.      Please find edited version of the abstract.

A genetically engineered chimeric virus crTMV-CP-PLRV composed of the crucifer-infecting TMV (crTMV) RNA and the potato leafroll virus (plrv) coat protein (CP) was obtained by agroinfiltration of Nicotiana benthamiana with the binary vector pCambia-crTMV-CPPLRV. The enormous quantity of the chimeric virus enabled not only for producing antibodies but also for direct visual observation of crTMV-CP-PLRV in the cell and to investigate its role in the infected plant. Localization of the crTMV-CP-PLRV in plant cells was examined by immunoblot techniques, light, and transmission electron microscopy. The distinguishing feature of the chimeric virus, the RNA virus with the positive genome, is localization in the nucleolus. The chimera transfers between vascular and nonvascular tissues. The chimera inoculum infects N. benthamiana mechanically. The role of the N-terminal sequence of the PLRV P3 coat protein in the cellular localization of the virus was inspected. We believe that the gene of the PLRV CP can be substituted with genes of other difficult to access plant pathogens to produce other helpful recombinant viruses.

CC-2.  English needs to be improved at various places in the manuscript.

R2.      Hopefully, it is corrected now.

CC-3.  Line 151: The authors that the CP is essential for the                                                            infectivity, however the data is missing. If authors are concluding                                             anything without the  ndetails of the results, it is hard for the readers to                              understand it well without a proper figure.

R3.      We constructed three binary vectors, in which gene of the PLRV CP was substituted with the gene of the original GFP and its two modifications (will be published elsewhere). We found that agroinoculation nicotiana with these vectors developed locally even after 7 days post-inoculation (Figure 2 added).

CC-4.  Figure 2, why authors used two different species for the WT and chimera                          virus infection? It is difficult to interpret the results if the two viruses are                           being infected in two different plants. Because we do not know if the                                difference is due to the virus or because of the two different plant species.

R4.      Please find corrected figures 1 and 2.

CC-5.  Figure 3: what is the difference in rows 1, 2 and 3? It should be labelled                              properly or mentioned in the figure legends/results. Are the rows 1, 2,                            and 3 same samples at different time points? If yes, then why there is                                     less staining in b-2 as compared to 1 and 3?

R5.      Images in rows 1, 2 and 3 show blots of petioles crosscuts of the different plants. This remark is added to the figure legend. 

CC-6.  Result section 3.4 is a bit complicated. Which vectors were constructed                             before and which one in this study? Not clearly mentioned. And if it is                           just the comparison of the chimeric virus in this study and the previous                             ones, it should be discussed with section 3.1. There is no need to make                          a separate section for this discussion.

R6.      Thank you for this comment. I replaced this section to the beginning of the paragraph “Results and Discussion.”

CC-7.  Figure 5: What is the difference in figure 5a and 5c? It is always a good                              idea to show comparison at the same scale in image so that the difference                      can be seen clearly. 5b is represented as an image of agro-inoculated N.                                 benthamiana petiole base. Can the authors provide the similar image at                            similar scale with non-infected N. benthamiana petiole base? It would                           improve the quality of results.                        

R7.      Please find corrections relevant to this figure (now Figure 4). Here images have close magnifications.

CC-8.  Figures 5b and 6b look exactly same. However, they have been described                        as uninfected and infected respectively. Is there some mistake? Similarly,                              figures 5a and 6a are exactly same, though they were taken at different                                  magnifications.

R8.      Corrected now. I beg your pardon. I did not succeed in checking out results of the automatic formatting system after submitting.

CC-9.  Section 3.7 describes the presence of NLS and NoLS signals in PLRV CP                                     which could be the reason for the localization of chimeric virus to                            nucleus. All these studies were done before as referenced by the                                             authors, so there is no point in discussing these again here. Authors can                              either remove this section or describe very briefly in section 3.6.

R9.      Corrected now. This sectioned was shortened.

CC-10. Likewise, section 3.8 is more kind of discussion, not results. So instead of                          discussing it in a separate section, I would recommend to include the                              content of this section in the corresponding result section. Also, the                                               sentence arrangement in this section is not thorough. It looks more like                            repetition of the sections described before in the manuscript.

R10.    The sections Results and Discussion are combined in this work. To follow your recommendation, I shortened section 3.8 of the manuscript removing most sentences with assumptions and speculations.            

CC-11. Line 30: luteoviruses has been introduced first time. It needs a bit more                              explanation.

R11.    At the very beginning of the manuscript added that PLRV belongs to the family Luteoviridae and is the cytoplasmic virus.

CC-12. We already know that the chimeric virus is infectious, so what do                            authors want to show new by the results in section 3.1.

R12.    Here we illustrate the infectiousness of this virus (Figures 1 and 2) that was absent in our previous publication.

CC-13. Line 221: what is normal CP?

R13.    Thanks, “normal” removed.

CC-14. Line 238: the introduction of N. benthamiana should be done earlier,                                   not in the result section.

R14.    N. benthamiana mentioned several times in Introduction and ‘Materials and Methods’. Here a description of the cellular structure of N. benthamiana precedes an analysis of the object on the subcellular level.

CC-15. References are not arranged properly.

R15.     Corrected now. I beg your pardon. I did not succeed in checking out results of the            automatic formatting system after submitting.

CC-16. Line 451: “We cannot also exclude the possibility that the gene of the                                  PLRV CP can be fused with the polyribonucleotides coding epitopes of                                   pathogenic bacterial, plant, animal, or human proteins to produce                                    vaccines.” This sentence is very broad and cannot be written without                                proper explanation, references and discussions. It would be best for the              manuscript if this conclusion is more focused.

R16.    Please find edited version of the section ‘Conclusions.’

Reviewer 4 Report

Your research showed the localization and effect of chimeric virus in plant organ.
Your original chimeric virus makes some of interests.

However, there is a vital mistake which makes me doubt the description accuracy.
Figs.5a and b is same as Figs. 6a and b, respectively.

You mentioned arginine rich sequence and the jelly roll structure.
Why you focus on and which data support this discussion?
I think you should discard Section 3.7.

Lots of information are lacked.
Fig.4 has no graph legend.
There is no information about plant growth condition, especially about potato.
the serial number of reference was missed.
MANY lanes were missed the format (miss-indention, miss-bolding, incorrect size).

Overall, you should resubmit after confirm and rewrite.

Author Response

Responses (R), critical comments (CC)

Dear Reviewer,             

Thank you for your careful review of our manuscript titled “The chimeric virus made from crTMV RNA and the coat protein of potato leafroll virus targeted to the nucleolus and infected N. benthamiana mechanically.”

I think the manuscript has been significantly improved after the recommended corrections.

CC-1.  owever, there is a vital mistake which makes me doubt the description                               accuracy. Figs.5a and b is same as Figs. 6a and b, respectively.

R1.      Corrected now. I beg your pardon. I did not succeed in checking out results of the automatic formatting system after submitting.

CC-2.  You mentioned arginine rich sequence and the jelly roll structure.                                        Why you focus on and which data support this discussion?

R2.      It is an interesting subject. Here I discarded this discussion. I hope that we will come back to this theme in the future. Nevertheless, coat proteins of several +RNA viruses have a jellyroll structure, and their N-end have NLS signal sequences.    

CC-3.  I think you should discard Section 3.7.

R3.      Thank you for the critical comment. This section was essentially shortened.

CC-4.  Lots of information are lacked. Fig.4 has no graph legend.

R4.      Please pay attention to lines 199-204.

CC-5.  There is no information about plant growth condition, especially about                                 potato.

R5.      Please find added lined 76-77.

CC-6.  the serial number of reference was missed.

R6.      Sorry, I did not succeed in checking out results of the automatic                             formatting system after submitting. Please find correct version of the issue                  REFERENCES.

CC-7.  MANY lanes were missed the format (miss-indention, miss-bolding,                                    incorrect size).

R7.      Thanks, please see previous reply.

Round 2

Reviewer 2 Report

Upon reviewing the manuscript, I believe it has been improved significantly.

At the end, I want to add the same comment as I did in my initial review; it might be out of the scope of the high-throughput journal.

Reviewer 4 Report

Your manuscript was improved so much.

Your discussion is better than before because it is in line with your observation results, such as nucleic localization.